# Spanish Validation of the Self-Perceived Food Literacy Scale: A Five-Factor Model Proposition

**DOI:** 10.3390/nu14142902

**Published:** 2022-07-15

**Authors:** Bárbara Luque, Joaquín Villaécija, Ana Ramallo, Margarida Gaspar de Matos, Rosario Castillo-Mayén, Esther Cuadrado, Carmen Tabernero

**Affiliations:** 1Maimonides Biomedical Research Institute of Cordoba (IMIBIC), 14004 Cordoba, Spain; z72viroj@uco.es (J.V.); rcmayen@uco.es (R.C.-M.); esther.cuadrado@uco.es (E.C.); 2Reina Sofia University Hospital, 14004 Cordoba, Spain; 3Department of Psychology, University of Cordoba, 14071 Cordoba, Spain; ana.rmallo7@gmail.com; 4Environmental Health Institute (ISAMB), University of Lisbon, 1649004 Lisbon, Portugal; margarida.gaspardematos@gmail.com; 5Instituto de Neurociencias de Castilla y León (INCYL), University of Salamanca, 37007 Salamanca, Spain

**Keywords:** food literacy, confirmatory factor analysis, internal consistency, healthy eating, young people

## Abstract

Food literacy is a combination of functional, critical, and relational skills that pave the way for navigating the food system properly, taking personally and contextually available resources into account. The aim was to validate the Spanish version of the self-perceived food literacy scale in university students to explore the factorial structure of it and to correlate food literacy with other variables. The sample was composed of 362 Spanish university students (314 women). The full questionnaire was administered online and also assessed adherence to a Mediterranean diet, impulsivity, and health-related quality of life for convergent validity testing purposes. Confirmatory factor analysis was conducted to determine the factor structure of the food literacy scale. The Spanish version of the scale showed good indices of internal consistency (Cronbach’s α = 0.894). Confirmatory factor analysis revealed a five-factor model that had a better fit index than the seven-factor model of the original scale. External validity was assessed by showing significant correlations with the rest of the variables. Therefore, the Spanish version of the scale is a reliable and valid measure of food literacy. It could be used to promote policies at Spanish universities to improve the food-related behaviors of students.

## 1. Introduction

The prevalence of coronary heart disease, overweight and obesity, type II diabetes, and cancer has led to a high level of social, media, and political coverage in recent years due to their close relationship with lifestyle [1,2]. In relation to the issue at hand, data on the increase in overweight and obesity are alarming. In Spain, it is estimated that 80% of men and 55% of women will be overweight or obese by 2030 [3].

The negative health impact of a sedentary lifestyle and a nutrient-poor diet occurs at any stage of life, although it is in the transitional stage between childhood and adulthood when the change in lifestyle is particularly significant, specifically regarding the way in which independence is experienced by university students when they begin their studies [4,5]. It is at this vital moment that most habits that will last into adulthood are established [6].

University students are in late adolescence, a period between the ages of 19 and 24 that marks the transition to adulthood in which they acquire important responsibilities. At this stage, there are changes in diet and the frequency of physical activity, which are related to weight gain in this population [6,7,8,9]. It has been determined that 20.6% of Spanish university students are overweight or obese (Spanish Nutrition Foundation [SNF], 2013).

During the university stage, responsibilities are accentuated, and new habits are acquired, such as shopping and cooking. This implies the emergence of new tasks such as considering what food is in the fridge or making a list of the food needed [4]. In addition to planning, many young people are responsible for meal preparation and cooking for the first time in their lives [4,7]. Several studies have indicated that the diet of university students is poor in whole grains, nuts, fruits, and vegetables and rich in sugary drinks, alcohol, red meat, and processed foods, high in fat, sugar, and sodium, and low in fibre [7,10,11].

### 1.1. Mediterranean Diet and Adherence in University Students

The habits of university students are far from being in line with recommendations that recognize the Mediterranean diet (typical of southern Europe and northern Africa) as a dietary pattern that protects against cardiovascular diseases [12] and is adequate in terms of quantity, nutritional quality, and energy. The Mediterranean diet is characterized by the frequent consumption of fruits and vegetables, nuts, whole grains, legumes, and spices. It is a plant-centred diet that leaves room for the consumption of moderate amounts of animal products and emphasizes tasty meal preparation rather than food restriction [13]. Studies support the benefits of adherence to this dietary pattern. On a physical level, it is related to a lower incidence of different types of cancer, diabetes, and cardiovascular diseases [14]; on a psychological level, it is positively related to greater perceived health, higher academic performance, and lower risk of suffering from clinical depressive symptomatology [15].

There is currently no consensus on adherence to the Mediterranean diet in university students; some studies point to moderate adherence [13,14,16], while others show low adherence [17,18]. In any case, what they do agree on is that patterns are increasingly moving away from nutritional recommendations and that adherence to the Mediterranean pattern may even be disappearing [15].

### 1.2. Behavior and Food Literacy

In recent years, concepts such as nutrition literacy and food literacy, both derived from health literacy, have been increasingly appearing in the literature due to the decline in personal competence in how to handle, access, and discriminate health-related information [19]. These terms arise as the basis of the social-ecological model that determines that eating behavior is determined by interacting individual, family, social-cultural, and physical environment factors [2,5,8].

Despite appearing similar, they are not. Nutritional literacy is defined as the set of cognitive, numerical, and basic knowledge skills that an individual must understand and use nutritional information [20]. It is understood as a form of health literacy applied to nutrition, for example, the ability to interpret a nutrition label [20]. However, food literacy refers not only to the skills needed to make healthy choices, but also to an understanding of the effect such choices have on health, the environment, and the economy [20]. It could be defined as the set of skills, knowledge, and practices that enable people to navigate through the food system in an effective, critical, and practical way to make better, healthier, and more sustainable choices, considering the resources available to them. Food literacy is based on four pillars: planning, selection, preparation, and consumption [21].

Planning refers to the prioritization of time and budget for food, i.e., planning in the formal and informal sense of what to eat regardless of changing circumstances and making balanced food choices based on available resources [21]. Planning among university students is rare. However, those who come from a healthier family background spend more time preparing and planning their meals. They also have better meal planning skills [2,5,8].

Food selection refers to accessing food by considering where it comes from, what it contains, how it is stored, the advantages and disadvantages of sources, and judging its quality [21]. This is influenced by factors such as accessibility in the university environment, the media, and peer pressure [2,5,8,22]. Seeing peers eating in a healthy manner facilitates healthier choices; conversely [8], seeing junk food advertisements on television increases the likelihood of junk food consumption [5].

Regarding food preparation, it should be understood as the ability to cook palatable food based on the available resources and equipment, considering appropriate hygiene and safety methods [21]. Therefore, the implications of the concept are not only mechanical, but also conceptual [23]. Wilson et al. [9] pointed out that the university population reports a generalized shortage of these competencies, and although they perceive themselves as being capable of carrying out the mechanical implications such as chopping food, they are more limited in terms of carrying out conceptual ones, such as planning all the resources and time necessary to cook several dishes concurrently such that these can be served at the same time.

Finally, the consumption of such food should involve awareness of the impact of eating on one’s own well-being and health, the conception of eating as a social event, and the satisfaction of nutritional needs and restrictions necessary for one’s own well-being based on one’s conditions or circumstances [21]. Several studies have indicated that boys are less interested than girls in learning about well-being and find nutritional recommendations unhelpful. However, neither boys nor girls tend to perceive the risk of developing chronic diseases by engaging in these eating habits, and they tend to be less aware than adults of the benefits of consuming, for example, fruits and vegetables [24].

These four pillars (planning, selection, preparation, and consumption) involve critical, functional, and relational competencies. Critical competencies refer to the appropriate discrimination between different cultural and social perceptions of production and consumption. Functional competencies are related to accessing, understanding, and using nutritional information. Relational competencies are based on appropriate interaction with system entities, with the individual being able to set goals and targets for health [20,25]. Balancing these competences in daily practice increases the likelihood of better decisions throughout the food system, contributing to consumer well-being [25].

### 1.3. Involvement of Other Psychological Variables

The integral nature of the concept of food literacy leads us to mention impulsivity, a multidimensional psychological variable that influences eating behavior. This is a behavior in which internal or external stimuli are acted upon with little or no consideration of the consequences involved, which may lead to dangerous or inconsistent actions [26,27,28]. Impulsivity has an impact on the nature and quantity of dietary intake [28,29], such that, in some cases, high levels of impulsivity are related to higher caloric and unhealthier intakes, especially when these options are readily accessible and available, as is the case for university students [27]. A recent study suggests that such impulsivity may increase the risk of pathological tissues such as night-time binge eating or obesity [30].

Finally, the importance of the impact that all the above has on the health-related quality of life (HRQoL) or perceived health of such students should be noted. This term, coined by the WHO [31], refers to the perception of the vital position that an individual occupies in the value system and cultural environment in which he or she lives according to the goals, expectations, values, and objectives that he or she has. Studies have found a positive relationship between adolescents’ perceived health and healthy food intake and vice versa [32]. In the opposite direction, and in terms of clinical symptomatology, studies by Gómez-Donoso et al. [33] and Pagliai et al. [34] have linked the consumption of soft drinks, added fats, and sauces to a greater likelihood of depressive symptoms.

Despite its importance, health promotion in the university life span has sometimes been overlooked [7]. Similarly, the impact of nutrition on the psychological level was neglected until a few decades ago, as it was only associated with personal, political, and socio-economic variables [35]. This idea is increasingly being banished, and much more holistic approaches to the issue are being taken. However, the term food literacy is relatively new and complex, and measurement tools are few, with limited psychometric properties [36].

For that reason, the first aim of this study was to adapt and validate the food literacy scale developed by Poelman et al. [29] to Spanish. Furthermore, the second aim was to explore the factor structure of the adapted scale and to establish correlations with the rest of the variables. Positive and significant relationships were hypothesized between food literacy, adherence to the Mediterranean diet, and perceived health, while negative and significant correlations were hypothesized between food literacy and impulsivity; that is, it is believed that a higher score in food literacy will be associated with greater adherence to the Mediterranean diet, higher perceived health, and lower impulsivity.

## 2. Materials and Methods

### 2.1. Participants

The sample consisted of 362 university students from different degrees and institutions. The age range was between 18 and 36 years (*M* = 22.36 years, *SD* = 3.76). Concerning the anthropometric data of the sample, the weight range was between 37 and 98 kg (*M_weight_* = 61.28 kg, *SD* = 10.46), the height range was between 150 and 190 cm (*M_height_* = 166.32 cm, *SD* = 7.17), and consequently the body mass index (BMI) range was between 14.45 and 33.33 (*M_BMI_* = 22.11, *SD* = 3.23). The sociodemographic data of the sample are shown in Table 1.

### 2.2. Instruments

The questionnaire consisted of five scales. Each of them is described below:

Sociodemographic and anthropometric data: descriptive characteristics of the sample were collected. The variables included were age, sex, university, degree, year of study, usual residence (family, friends, partner, flatmate, student residence, or other), and work activity (yes/no, part-time/full-time). Anthropometric data such as weight (kg) and height (cm) were self-reported, and considering both scores, the BMI was calculated. Based on the World Health Organization classification, participants were grouped according to their BMI score as follows: underweight (<18.5 kg/m^2^, *n* = 36), normal weight (18.5–24.9 kg/m^2^, *n* = 268), overweight (25–29.9 kg/m^2^, *n* = 48) and obesity (>30 kg/m^2^, *n* = 10).

Self-perceived food literacy (SPFL) scale: a translation of the Dutch scale designed and validated to measure food literacy [29] in adult populations was carried out. The scale is composed of a total of 29 items grouped into eight dimensions (food preparation skills, resilience, and willpower, type of healthy snacks, social and mindful consumption, observing nutrition labels, daily planning, health budget, and healthy food storage). The response scale is Likert-type and ranges from 1 = never, to 5 = always. The scoring of the reverse items was corrected so that a higher score implied higher food literacy. 

Mediterranean diet adherence screener (MEDAS): a questionnaire on adherence to the Mediterranean diet validated in Spanish [37] was administered. It is composed of 14 items, 12 of which are related to the consumption of foods belonging to the Mediterranean diet (e.g., olive oil, vegetables, fruits, wine) and two of which refer to cooking and eating preferences (fats used and type of meat). The items are written in a dichotomous way (yes/no). Each item is scored with 0 or 1 point. Scores below nine are considered to indicate low adherence to the Mediterranean dietary pattern.

Impulsivity (negative urgency, lack of premeditation, lack of perseverance, sensation seeking, and positive urgency; UPPS-P): the short version of the UPPS-P [38] measuring impulsivity was administered. It was validated in a Spanish university population [26]. The UPPS-P (short version) consists of 20 items grouped into five factors (positive urgency, negative urgency, lack of premeditation, lack of perseverance, and sensation seeking). The items are presented with a Likert-type scale with four response options ranging from 1 = strongly agree, to 4 = strongly disagree.

Health questionnaire (SF12 version 2): this instrument is the result of the Spanish adaptation by Alonso et al. [39] of the SF12 health questionnaire [40,41,42]. It is composed of 12 items from eight dimensions of the SF36 health questionnaire (physical function, social function, physical role, emotional role, mental health, road, bodily pain, and general health). The response options are dichotomous for four items, and the rest are presented with a three- or five-point Likert-type scale. The final score was calculated with a summatory where higher scores were indicative of better physical and mental health status.

### 2.3. Procedure

This was an instrumental, descriptive, and cross-sectional study. The project was approved by the Bioethics Committee of our institution (blinded for peer review). After translation and back-translation of the self-perceived food literacy (SPFL) scale, it was grouped with four other instruments, and an online questionnaire was created. The tool was piloted on a sample of 14 university students who met the inclusion criteria. No modifications were made after this. These participants were then excluded from further data collection. The aims of the study were explained and informed consent for participation was requested at the beginning of the questionnaire throughout the fieldwork. Data collection was carried out from 10 November 2020 to 25 January 2021 via the Moodle platform and via the social network Instagram. Participants did not receive any incentives.

### 2.4. Design and Data Analysis

Firstly, descriptive statistics were calculated for the sample (socio-demographic and anthropometric data). Secondly, a confirmatory factor analysis (CFA) was carried out to check the structure of the questionnaire, as research indicates that this is appropriate for cases of translated scales that have been previously validated. Maximum likelihood estimation was used. As for the thresholds of the calculated estimates, levels between ≥0.95 and <0.97 for CFI, levels between ≥0.90 and ≤0.95 for GFI and TLI, and levels between ≥0.85 and ≤0.90 for AGFI indicate an acceptable model fit while levels ≤0.05 for RMSEA indicate a good fit. The lowest AIC value is considered to indicate the best model [43]. Thirdly, the internal consistency using Cronbach’s alpha was calculated to determine the reliability of the scale. Finally, given that a non-normal distribution was confirmed for all the study variables, convergent validity was calculated through Spearman correlations between the SPFL scale and the other variables (MEDAS, UPPS-P, and SF12 version 2). All analyses were carried out with the statistical data package SPSS (Statistical Package for Social Sciences) version 25 (SPSS Inc., Chicago, IL, USA) and Amos version 25. Gpower 3.1 was used to verify that the achieved statistical power was at least 0.8 for all analyses with the actual sample size.

## 3. Results

To understand the factor structure of the food literacy scale adapted to Spanish, four different models were analyzed. The fit indices of the models are shown in Table 2.

First, a one-factor model (M0) represented by the 29 items proposed by the authors of the original scale was analyzed. M0 was discarded since the three items (items 17–19) corresponding to factor 4 (consideration of eating as a social and conscious act) of the original scale [29] were not statistically significant and were subsequently removed. After removing the non-significant items, a second model (M1), also unifactorial (26 items), was analyzed without satisfactory results (see Table 3).

The third model (M2) that was analyzed consisted of 25 items. It was a model of seven interrelated factors like the one presented by the authors of the original scale. In this model, factor 3 (items 13–16) was removed due to the low reliability reported by Poelman et al. [29] (Figure 1). M0 and M1 were thus discarded due to their poor fit compared to M2. Finally, a fourth model (M3) consisting of five interrelated factors was analyzed because of the low significance of items 18 and 19 after analyzing M2. Due to the impossibility of a factor being represented by only one item, the remaining item was also discarded (item 17) and the factor was eliminated in this model. Although the fit between M2 and M3 was quite similar, M3 showed the best fit after carrying out further analyses. Its internal consistency was adequate (α = 0.81).

The five factors obtained from M3 were named: cooking skills (items 1–6), emotional management (items 7–12), healthy consumption as a priority (13–16, 24, and 25), nutritional literacy and planning (20–23), and availability of ultra-processed foods (26–29). Cronbach’s alpha for each factor was 0.83, 0.71, 0.84, 0.82, and 0.82, respectively. As shown in Table 3, the concurrent validity analyses conducted showed a positive and significant relationship between food literacy and adherence to the Mediterranean diet and perceived health, and a negative and significant relationship between food literacy and impulsivity.

To assess the differences between each BMI group and the factors of the SPFL, a repeated measures Analysis of Variance (ANOVA) was carried out. The results indicated that the multivariate test was significant, traza pillai = 0.08, *F*(12,1071) = 2.51, *p* = 0.003, η^2^_p_ = 0.03, observed potency = 0.976. The within-subject effects test indicated that the interaction between the factors of the SPFL and the BMI groups was also significant, *F*(12,1432) = 2.35, *p* = 0.005, η^2^_p_ = 0.02, observed potency = 0.967. A multivariate ANOVA indicated that the multivariate test for the BMI group was also significant, traza pillai = 0.09, *F*(15,1068) = 2.28, *p* = 0.004, η^2^_p_ = 0.03, observed potency = 0.983. The between-factor effect test showed that factor 4 was significant, *F*(3,358) = 3.75, *p* = 0.011, η^2^_p_ = 0.03, observed potency = 0.809. However, DMS post hoc tests also showed differences in factor 1 between underweight (*M* = 3.48) and obesity (*M* = 4.15) groups, *p* = 0.026, and in factor 4 between normal weight (*M* = 3.26) and both underweight (*M* = 2.76) and overweight (*M* = 2.91) groups (*p* = 0.005 and *p* = 0.28, respectively). Also, a difference was found in factor 5 between underweight (*M* = 3.46) and normal weight (*M* = 3.95) groups, *p* = 0.006 (Figure 2).

## 4. Discussion

Food literacy is a recent multidimensional construct that lays the foundation for navigating the food system appropriately, according to the circumstances and resources available [21]. The aim of this study was to explore the psychometric properties of the Spanish adaptation of the food literacy scale developed by Poelman et al. [29] for its subsequent validation. The results show that it is a valid and reliable scale for measuring this construct in a university population. The scale is composed of 16 items grouped into five factors.

Previous research in other countries has explored the factor structure of food literacy [44,45]. However, although they have the same conceptual and definitional framework as a reference, the distribution and representation of the subcomponents are different between them [46]. In this study, the elimination of the factor that considers food as a social act due to its low significance could be explained because the population included university students and not adults with whom Poelman et al. [29] developed their scale. University students perceive time constraints as a barrier to implementing healthy eating behaviors [22]. If they perceive it as being difficult to dedicate time to healthy eating at the same time as, for example, attending classes [47], it is understandable that they also do not attach much importance to the social gathering around food in their daily lives. in any case, none of these studies have made an empirical observation based on the life experience of the target population [46] but have rather relied on experts for the development of the scale or followed nutritional recommendations from the country, or based on previous research [44,48]. This underestimates the influence of geography and culture on the concept [49] and demonstrates psychometric differences between scales across countries.

The differences between M2 and M3 were mainly in the distribution of the items and the reliability indices as well as the individual factors. M3 includes subcomponents in the same factor that Poelman et al. [29] included separately. This change in the distribution and number of items and the consequent impact it had on the total and factor reliability of the scale may be due to the interrelated nature of each of the components of food literacy, even though theoretically the authors Vidgen and Gallegos [21] grouped and classified them separately [46]. This could be due to the leap between theory and practice, typical of the beginnings of measuring a new construct. Although the differences between M2 and M3 were minimal, the latter still represented seven of the subcomponents proposed by the theoretical authors and had a total scale reliability (α = 0.89), higher than that shown in previous studies [44,48].

On the other hand, external validity was demonstrated through the relationship between food literacy and other variables. As expected, those with higher adherence to the Mediterranean diet indicated better food literacy, which also occurred with all the factors. This is consistent with previous research that found a relationship between food literacy components (planning, food knowledge, cooking skills, etc.) and adherence to the Mediterranean diet. These components are conceived as variables that favor the frequency of fruit and vegetable consumption and therefore facilitate adherence [48]. This result offers an opportunity for educational intervention, not only for university students, but also for the people who are responsible for this task of self- and hetero-care, in cases where they are not yet living independently [45].

In terms of impulsivity, the results showed the expected correlation with the full scale and its factors: those who were more impulsive scored lower in food literacy. These results are consistent with prior research. For example, in relation to factor 2 (emotional management), this is in line with previous studies [50], which highlighted the importance of the emotional state and its management in relation to healthy eating behaviors. The correlation between factor 2 and factor 5 (availability for ultra-processed foods) is related to studies indicating that, high consumption of ultra-processed foods is associated with increased impulsivity when eating, which in turn correlates with depressive or anxious states [51]. The negative correlation between impulsivity and factor 4 (nutritional literacy and planning) also seems coherent since food literacy is positively related to self-control [29]. This pattern of correlations is especially important in university students, as in the study conducted by Hernandez et al. [52], 51% of the sample admitted to eating less healthily during the university exam period. In addition, those with higher food literacy scores were found to perceive themselves as healthier. This result was found for the full scale and for the five factors. This positive relationship between food literacy, health, and well-being has also been found in other studies [25,48]. Regarding differences between BMI groups on SPFL factors, the fact that participants with a normal weight showed better levels of nutritional literacy and planning than underweight and overweight participants may indicate that better knowledge in these aspects of nutrition is beneficial for a healthier weight. This result is in line with the study by Wijayaratne et al., where a negative association between BMI and food literacy was found, along with a positive effect of food literacy on preparing healthy meals [45]. In addition, the lower availability of ultra-processed foods in the underweight group compared to the normal weight group could be indicative of a type of dietary restraint in the former group. More research is necessary to explore these relationships.

Despite its contributions, this study has limitations. Regarding the size of the sample, although it was adequate for the development and validation of the scale, it may not be representative of the Spanish university population. Specifically, one of the limitations in relation to our sample is the low representation of men. It would be necessary, in future studies, for the sample size to be larger and to include homogeneous and gender-balanced groups from different programs of study and regions. On the other hand, the existing psychometric limitations are the result of the novelty of the concept. It would be interesting to develop a scale in Spanish that includes the 11 subcomponents of food literacy instead of looking for an adjustment to the four major domains (planning, selection, preparation, and consumption) where, according to Amouzandeh et al. [46], some bias could lie. A longitudinal study could also be conducted, including objective concept measures in addition to self-reported ones.

## 5. Conclusions

To our knowledge, this study presents the first food literacy measurement tool that has been validated in Spanish in a university population. Although it is still a wide field to be explored, it is now possible to understand the relationship between food literacy and adherence to a healthy and sustainable eating pattern (Mediterranean diet) and how this is related to intrapersonal variables such as impulsivity. But, above all, the results allow us to establish the relationship that these competencies have on the perceived health of university students. Therefore, it would be advisable to encourage the development of food literacy to increase adherence to healthy eating patterns and to promote the physical and psychological health that derive from it. The Spanish university system could and should be one of the central channels for the acquisition and maintenance of the primary base knowledge that guarantees well-being regarding food intake.

## Figures and Tables

**Figure 1 nutrients-14-02902-f001:**
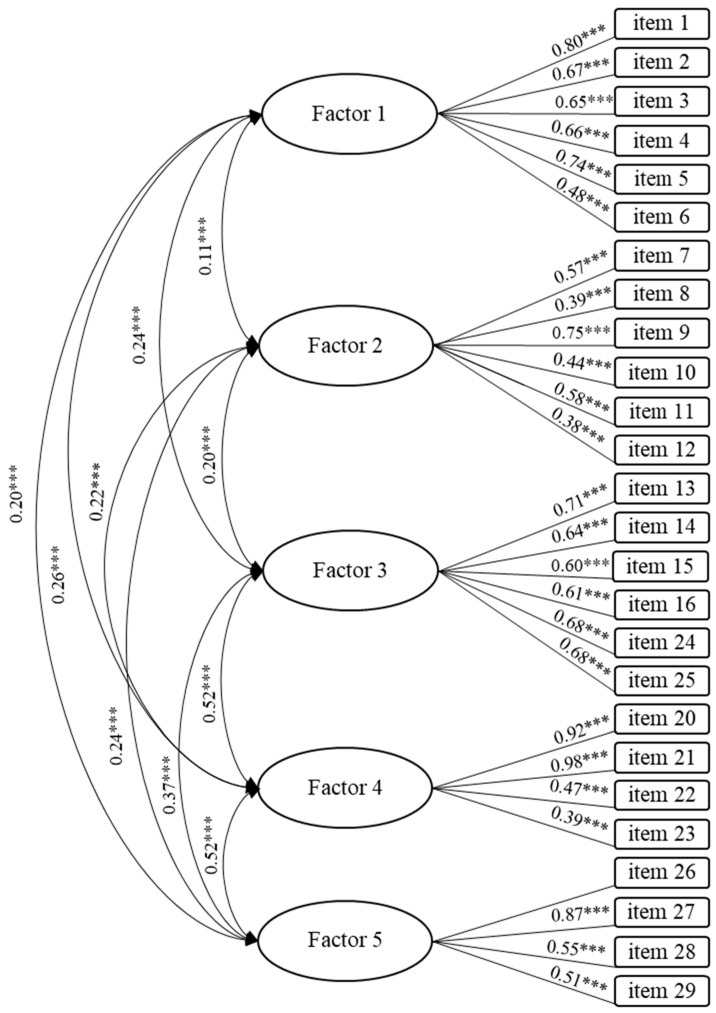
Five-factor model (Model 3 with 26 items). F1 = cooking skills; F2 = emotional management; F3 = healthy consumption as a priority; F4 = nutritional literacy and planning; F5 = availability of ultra-processed foods (*** *p* < 0.01).

**Figure 2 nutrients-14-02902-f002:**
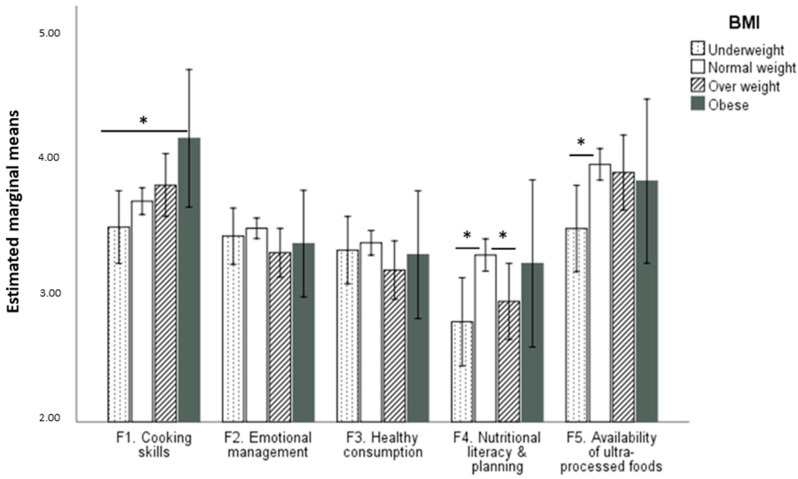
Estimated marginal means and post hoc significant differences for each BMI group in the SPFL factors (* *p* < 0.05).

**Table 1 nutrients-14-02902-t001:** Percentages of the total sample according to sex, current work activity, housing, cohabitation, university, degree, area of knowledge, and course.

	*%*	*N*		*%*	*N*
Sex			University (region)		
Male	13	47	Cataluña	3.6	13
Female	86.7	314	Madrid	8.01	29
Not answered	0.3	1	Comunidad Valenciana	4.14	15
Current work activity			Castilla y León	2.5	9
No	74.9	271	Andalucía	77.35	280
Yes	25.1	91	Galicia	2	7
-Full time	50	45.5	País Vasco	2.4	9
-Part time	50	45.5			
Family residence			Knowledge area		
Yes	59.7	216	Scientific/Technological	14.91	54
No	40.3	146	Health Sciences	54.14	196
Living in			Social Sciences	24.85	90
Family	54.4	197	Humanities	6.1	22
Friends	10.8	39			
Couple	9.7	35	Course		
Student residence	1.7	6	First	18.8	68
Roommates	19.3	70	Second	47.4	171
Other	4.1	15	Third	6.9	25
Qualification			Fourth	15	54
Degree	87.84	318	Fifth	7.4	27
Master’s Degree	12.16	44	Sixth	4.4	16

**Table 2 nutrients-14-02902-t002:** Fit indices of the CFA of the four models of the Spanish adaptation of the SPFL scale.

Model	χ^2^	gl	*p*	CFI	RMSEA	LO 90%	HI 90%	GFI	AGFI	TLI	AIC
M0	2463.03	377	<0.001	0.50	0.124	0.119	0.128	0.66	0.61	0.46	2579.03
M1	2252.01	299	<0.001	0.52	0.135	0.129	0.140	0.65	0.59	0.48	2356.01
M2	466.80	254	<0.001	0.94	0.048	0.041	0.055	0.91	0.88	0.91	608.80
M3	487.58	284	<0.001	0.95	0.045	0.038	0.051	0.91	0.87	0.94	621.58

CFA: Confirmatory Factor Analysis; CFI: Comparative Fit Index; RMSEA: Root Mean Square Error of Approximation; GFI: Goodness of Fit Index; AGFI: Adjusted Goodness of Fit Index; TLI: Tucker Lewis Index; AIC: Akaike Information Criterion.

**Table 3 nutrients-14-02902-t003:** Correlations (Spearman) between the Mediterranean diet adherence screener (MEDAS) with the total score of the self-perceived food literacy (SPFL), the impulsivity (UPPS-P short version), the self-perceived health questionnaire (SF12), and the five factors of food literacy (F1 = cooking skills; F2 = emotional management; F3 = healthy consumption as a priority; F4 = nutritional literacy and planning; F5 = availability of ultra-processed foods). Data are presented as: *Mn* = median, *SD* = standard deviation, *S* = Skewness; *K* = Kurtosis; *K*-*S* = Kolmogorov–Smirnov.

	MEDAS	SPFL	UPPS-P	SF12	BMI	F1	F2	F3	F4	F5	*Range*	*Mn*	*SD*	*S*	*K*	*K*-*S*
MEDAS	1										1–12	8.00	1.89	−0.34	0.29	0.13 **
SPFL	0.59 **	1									1.93–4.79	3.50	0.55	−0.21	−0.46	0.05 *
UPPS-P	−0.10	−0.26 **	1								1.25–3.50	2.20	0.40	0.16	−0.01	0.05 *
SF12	0.20 **	0.31 **	−0.26 **	1							14–46	35.00	5.83	−0.57	−0.27	0.10 **
BMI	−0.03	−0.04	0.03	−0.13 *	1						14.5–33.3	21.51	3.23	0.90	0.84	0.09 **
F1	0.45 **	0.71 **	−0.16 *	0.20 **	0.07	1					1–5	3.83	0.61	−0.59	−0.08	0.09 **
F2	0.34 **	0.65 **	−0.30 **	0.33 **	−0.10	0.36 **	1				1.33–5	3.50	0.43	−0.26	0.05	0.07 **
F3	0.59 **	0.82 **	−0.14 *	0.23 **	−0.12 *	0.51 **	0.43 **	1			1.17–5	3.33	0.78	−0.25	−0.25	0.08 **
F4	0.40 **	0.72 **	−0.20 **	0.19 **	0.01	0.35 **	0.31 **	0.55 **	1		1–5	3.25	1.03	−0.22	−0.63	0.08 **
F5	0.33 **	0.63 **	−0.12 *	0.18 **	0.00	0.24 **	0.34 **	0.42 **	−0.34 **	1	1–5	4.00	1.01	−0.98	0.44	0.15 **

* *p* <0.05, ** *p* <0.01.

## Data Availability

Not applicable.

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
