# Peer review of "Spanish Validation of the Self-Perceived Food Literacy Scale: A Five-Factor Model Proposition"

_nutrients, 2022, doi:10.3390/nu14142902_

Round 1

Reviewer 1 Report

On one side the Spanish version of food literacy interecting, one other side the eating habits of students, depend on many factors, and getting more and more internationl. The healthy food idea is not very positive in my understanding, all food are healthy, nobody is selling unhealthy food, the diet and the lifestyle can be different, and unhealthy too. The Mediterrian diet is underlined in the papaer, but not really described, and to separated from the street food, which is one of the main food of several students.

Author Response

The authors would like to thank the reviewer for the positive assessments of the presented work. In relation to the suggestion made about the Mediterranean diet, the title of section 1.1 has been modified to emphasize that this diet is discussed here (now called "Mediterranean diet and adherence in university students"). This section also describes the Mediterranean diet as the reviewer indicates it should be.

The habits of university students are far from being in line with recommendations that recognize the Mediterranean diet (typical of southern Europe and northern Africa) as a dietary pattern that protects against cardiovascular diseases [12] and is adequate in terms of quantity, nutritional quality, and energy. The Mediterranean diet is characterized by the frequent consumption of fruits and vegetables, nuts, whole grains, legumes, and spices. It is a plant-centred diet that leaves room for the consumption of moderate amounts of animal products and emphasizes tasty meal preparation rather than food restriction [13]. Studies support the benefits of adherence to this dietary pattern. On a physical level, it is related to a lower incidence of different types of cancer, diabetes, and cardiovascular diseases [14]; on a psychological level, it is positively related to greater perceived health, higher academic performance, and lower risk of suffering from clinical depressive symptomatology [15].” (Lines 65 - 82)

Reviewer 2 Report

MAJOR REMARKS

- The rationale to consider specifically impulsivity for discriminant validity is unclear. Please clarify this in the text.

- It is also unclear which criteria were used to assess convergent and discriminant validity. Significant positive and negative correlations are not sufficient, since several of those correlations are very weak, and only significant due to the sample's size.

- The prior issue must be clearly discussed, as the results presented do not indicate adequate discriminant validity if impulsivity is used for that.

- In the statistical analysis, report how was normality assessed. Non-normal variables should not be presented as means and standard deviations.

- Also, refer in the methods which correlation coefficient was used. If variables do not present normal distribution (including scales' scores), Spearman's correlations should be used instead.

- Despite in part this may be due to the number of items, the fact that Cronbach's alpha for the total is higher than for all factors should be discussed.

- The discussion should also focus on the correlations with the factors, not only with the overall scale.

MINOR REMARKS

- Please rephrase sentence in lines 25-27, as it is unclear.

- Line 68: replace "calories" with "energy".

- Please rename section 1.3, as it does not refer only to impulsivity.

- Instead of weight and height, please report BMI (Table 1). I also suggest to present and discuss the correlations of SPFL (total and factors) with BMI.

- There are only 2 sexes in our species, so please correct data presentation in table 1 (1 participant with "other" sex). In line with this, correct "gender" to "sex" in line 331.

- Lines 183-4: were weight and height self-reported?

- For all scales (lines 185-211) indicate the range of values for the total. The same to the 5 factors of SPFL.

- Indicate how many participants were included in the pilot (lines 216-7).

- In the limitations, refer explicitly the low number/proportion of males.

Round 2

Reviewer 2 Report

1. In response to the comment “The rationale to consider specifically impulsivity for discriminant validity is unclear. Please clarify this in the text.” the authors have “revised the corresponding discussion paragraph and expanded its justification”. Please add the main idea regarding impulsivity being used to assess discriminant validity in the methods section.

2. Regarding the comment “It is also unclear which criteria were used to assess convergent and discriminant validity. Significant positive and negative correlations are not sufficient, since several of those correlations are very weak, and only significant due to the sample's size.”, the changes do not correspond to what was meant. Skewness, kurtosis and Kolmogorov-Smirnov test have nothing to do with it! The authors should explain the criteria, i.e., besides being significant, which correlation value was considered the minimum to consider there was convergent validity?

3. The statistical analysis used should be described in section 2.4, not in the results. The authors do not have to present the assessment of normality, but to perform it correctly and describe how it was assessed. Kolmogorov-Smirnov’s test is not adequate for large samples, so please indicate only skewness and kurtosis (and none of these needs to be presented in table 3).

4. Did age, height, weight and BMI present a normal distribution? If not, correct the use of means and SD.

5. Comment “Despite in part this may be due to the number of items, the fact that Cronbach's alpha for the total is higher than for all factors should be discussed.” » Reply: “Thanks to your input we have reviewed the reliability analyses and detected an error. After correcting the error the reliability of the entire scale is within the expected range. The reliability of the overall scale is .81 which is within the range of the 5-factor reliability”. The issue remains... Please discuss the lower alpha for factor 2.

6. In the repeated measures ANOVA text, replace “traza pillai” with “Pillai’s trace”. Also, please consider in this analysis the issue of the normality of BMI. Moreover, the non-normality of data must be addressed, considering this analysis. This also aplies to the MANOVA. Please add these analysis to subsection 2.4.